# Does Pre-Service Teacher Preparation Affect Students' Academic Performance? Evidence from China

Xinqiao Liu [1], Wenjuan Gao [2] and Luxi Chen [3,*]

1   School of Education, Tianjin University, Tianjin 300072, China
2   Institute of Higher Education, Beihang University, Beijing 100191, China
3   Faculty of Education, Beijing Normal University, Beijing 100875, China
*   Correspondence: luxichen@bnu.edu.cn

**Abstract:** Pre-service teacher preparation (PSTP) is generally considered a significant predictor of student achievements. This paper adopted a multi-tier linear model to estimate the PSTP effects on student performance by taking teachers and students in the high schools of Haidian District, Beijing, China, as the research population. It used exploratory factor analysis to classify PSTP into two categories: content knowledge preparation and pedagogical content knowledge preparation; and described the status of PSTP in three subjects: Chinese, mathematics, and chemistry. The study found differences in PSTP by subject. In Chinese, teachers' content knowledge preparation significantly negatively affected student performance, and their pedagogical content knowledge preparation significantly positively influenced student performance. In mathematics, PSTP had no significant effect on student performance. In chemistry, teachers' pedagogical content knowledge preparation had a significantly negative effect on student performance. Based on the findings of the empirical study, the study proposes further identifying PSTP's role in student performance by subject, strengthening the focus on pre-service preparation skills in recruiting Chinese and chemistry teachers, and developing a more suitable system for teacher selection and training.

**Keywords:** pre-service; teacher preparation; academic performance; high school student; college entrance examination





## 1. Introduction

Teachers are a crucial factor in improving the quality of education and students' academic performance. To this end, governments are committed to enhancing teachers' quality. Den Brok et al. (2004) reported that 7–15% of the variance in student achievements could be attributed to differences between teachers [1]. Day et al. (2006) argued that the differences between teachers might explain 15–30% of the variance in student achievements [2]. The research by Hanushek and Rivkin (2010) has referred to two generally accepted findings regarding teachers' contributions to student performance. First, there is considerable variation in the quality of teachers as measured by the value of increased achievements, future academic achievements, or earnings; second, variables commonly used to determine careers and salaries, such as academic qualifications and certificates, do not explain the variations in measured teacher quality. Therefore, the teacher characteristics we observed are not representative of the quality [3]. Through an empirical study, Rivkin et al. (2005) stated that the variation in teacher quality is rarely explained by observable characteristics [4] and that the observable characteristics of teachers could shed light on only about 5% of the variation in student achievements [3].

Most studies have been conducted on the effects of teachers on students' academic performance by their attributes and characteristics (e.g., gender, teaching standing, academic qualifications, etc.) [5–7], but it remains insufficient to study the effects of teachers' professional development, such as pre-service teacher preparation (PSTP), on students' academic performance. PSTP, also known as initial teacher training, means the preparation

of teachers for theoretical knowledge and teaching before they start their teaching [8]. PSTP aims to help them transition from campus to career. The No Child Left Behind Act proposed that every student could be taught by highly qualified teachers. Consequently, the US government has invested heavily in teachers' professional preparation and career development to better PSTP and highlight its importance. Countries have focused on PSTP from a variety of perspectives. Through comparison, Zhu Xiaohu and Zhang Minxuan (2017) unveiled that Finland has great strengths in pre-service education and improves teachers' professionalism through rigorous selection. Shanghai, China, has invested heavily in induction training and others to enhance teachers' professionalism through standardized training [9–11]. In addition, Brazil's Accelerated Learning Program trains new teachers through a highly structured curriculum [12]. Schools hope to attract outstanding talents with teachers' qualities to their schools. By improving the PSTP model, they can refine the quality of teachers they recruit and boost their sustainable development and students' academic performance [13]. Ramírez (2006) found that well-prepared math teachers could teach more mathematical content in class [14]. Akiba (2011) noted that teacher preparation for diversity reported by pre-service teachers is associated with positive changes in pre-service teachers' beliefs about diversity in their personal and professional environments [15]. Little and Anderson (2016) found that although most pre-service teachers think that their beliefs are compatible with problem-solving tasks, the ability of middle school students, preparation time, and cooperative teachers are the main factors affecting their performance on problem-solving tasks [16]. Shaukat and Chowdhur (2021) analyzed the perceptions of 52 Australian and 68 Pakistani pre-service teachers (PST) on the professional standards for teachers to compare teacher preparation in the two countries, and concluded that standards-based integrated teacher preparation programs are more effective than non-integrated teacher preparation programs in promoting professional skills and competency development [17]. Evagorou et al. (2015) compared the teacher preparation courses in England, Finland, France, and Cyprus, and found that the pre-service teacher training programs in the different countries have different focuses. Finland attaches great importance to teacher preparation and encourages teachers to enhance their teaching performance through research skills. Meanwhile, Finnish university-affiliated schools and specially trained tutors emphasize exercises during preparation. Training in Cyprus and the UK also includes research training for students, but the focus is less pronounced than in the Finnish system [18].

There is a strong correlation between PSTP knowledge and content knowledge in teaching. Shulman (1986) summarized content knowledge in teaching into three core aspects: subject matter content knowledge, pedagogical content knowledge, and curricular knowledge [19]. However, the subject matter knowledge that teachers learn in higher education is not directly applied to primary and secondary school instruction, so it is inconclusive whether teachers' content knowledge preparation is beneficial to student performance. Based on the data from the Longitudinal Study of American Youth (LSAY), Monk (1994) showed that teachers' insight into what they have learned has a positive impact on student performance [20]. According to the analysis of Rowan et al. (1997), teachers' subject knowledge and expectations directly influence students' mathematics performance. These effects depend on the students' average ability in a given school [21]. Hill et al. (2015) examined whether and how mathematics teachers' pedagogical content knowledge contribute to students' mathematics performance. After controlling for key covariates of students and teachers, teachers' mathematics knowledge was significantly associated with student performance in both first and third grades [22]. However, Eberts and Stone (1984) found no relationship between college-level mathematics courses and fourth-grade test scores [23]. In addition, content knowledge in teaching is an important determinant influencing learning gain and motivation development [24]. Baumert et al. (2010) explored the importance of teachers' content knowledge and content knowledge in teaching for high-quality mathematics teaching and students' progress in secondary schools, confirming the correlation between specific teachers' expertise and high-quality

teaching and students' learning [25]. Slavíčková (2020) uncovered a strong correlation between preparatory mathematics teachers' capabilities for applied digital technology and their activities in the curriculum [26]. Creativity-oriented tasks are integrated into university courses in mathematics, which can better allow pre-service teachers to develop innovative mathematical skills for future students [27]. Corcoran and Flaherty (2018) found no significant relationship between personality traits and the outcome variable, teaching performance. However, teaching performance in the past has also emerged as an important predictor of teaching performance apart from academic performance [28]. Furthermore, for under-resourced teaching, teachers should receive specialized training on adapting their curriculum plans for students with different ability levels [29]. Tunjera and Chigona (2020) recommend the adoption of a technology integration framework and pedagogical theory at the level of policy development in pre-service teacher training institutions [30].

Although previous studies agree that PSTP can play a part in students' academic performance, PSTP's role needs to be discussed by discipline given the significant variation in course content and teaching design across disciplines [31]. There is little research on the disciplinary differences in PSTP. Consequently, this study presents the following hypotheses based on the key research issues of existing PSTP studies.

**Hypothesis I.** *PSTP can be divided into one for general subject matter knowledge and one for diverse pedagogical content knowledge.*

**Hypothesis II.** *There is interdisciplinary heterogeneity in PSTP levels due to various learning approaches across disciplines.*

**Hypothesis III.** *PSTP can have significantly positive effects on student performance.*

## 2. Methods

### 2.1. Participants

The dataset for this study combines the data concerning students and teachers. Student data mainly refer to the standardized test scores of Chinese, mathematics, and chemistry for students who took the college entrance examination in Haidian District, Beijing from 2016 to 2019, including the first simulated test results of the college entrance examination and the results of the senior high school entrance examination. Specifically, the senior high school entrance exam scores represent the knowledge acquired by students before entering high school, i.e., the entrance scores of high schools. The first simulated test scores of the college entrance exam serve as a proxy variable for college entrance exam scores and represent the exit scores of high schools after students have experienced three years of learning and training in a high school.

Teacher data are derived from the Regional Teaching and Research Survey questionnaire conducted between February and March 2019 for schools in Haidian District, Beijing. The survey aims to understand teachers' demands for professional development and their needs for teaching and research in the new era of educational reform. Based on the globally used questionnaire derived from the TALIS (Teaching and Learning International Survey) regarding teachers' professional development, effectiveness, teaching practices, and classroom behaviors, the questionnaire was developed concerning teachers' current professional development in China [32].

Based on student data, this study matched teacher data with student data according to the names and schools of teachers participating in the survey, and the names and schools of teachers in the student data. Thus, the dataset can be deemed as a combination of both secondary and primary data. The matching resulted in the creation of a multi-tier database containing the scores of the high school entrance examination, the first simulated test of the college entrance examination, and the survey data of corresponding teachers and school names. Through matching, the 542 teacher data from 60 ordinary high schools were linked with the 39,894 student data. The number of students with Chinese, mathematics, and

chemistry scores was 14,296, 15,662, and 9936, respectively; and the number of Chinese, mathematics, and chemistry teachers was 195, 216, and 131, respectively [32].

*2.2. Measures*

Student data were measured using objective test scores. Students' scores on the first simulated test for the college entrance examination and the senior high school entrance examination were continuous variables. To compare the data across years, this study standardized the scores of each exam in the entire Haidian District based on students' graduation year, liberal arts and sciences, and types of exams.

PSTP was measured using a scale consisting of 10 questions covering multiple aspects of PSTP, including subject matter knowledge, teaching competencies, pedagogy, and student management. The question on the PSTP scale in the teacher questionnaire is, "Does the specialized course you have taken include the following? If so, do you think you are well prepared when you graduate?" The specific scale is shown in Table 1, which collects information on how teachers who are already in service feel about the relevant pre-service training before their employment. All questions are measured using a 5-point scale. In the empirical analysis, "Not included" is assigned a value of 1, "Inclusion; no preparation" 2, "Inclusion; preparation to some degree" 3, "Inclusion; well prepared" 4, and "Inclusion; very well prepared" 5. Thus, each question is transformed into a fixed interval variable.

**Table 1.** Pre-Service Teacher Preparation Scale.

| No. | Question | Not Included | Inclusion; No Preparation | Inclusion; Preparation to Some Degree | Inclusion; Well Prepared | Inclusion; Very Well Prepared |
|---|---|---|---|---|---|---|
| Q1 | Knowledge and understanding of subject areas taught | 1 | 2 | 3 | 4 | 5 |
| Q2 | Teaching ability of subjects taught | 1 | 2 | 3 | 4 | 5 |
| Q3 | General education and teaching method | 1 | 2 | 3 | 4 | 5 |
| Q4 | Teaching methods for subjects taught | 1 | 2 | 3 | 4 | 5 |
| Q5 | Tiered teaching for students with different abilities | 1 | 2 | 3 | 4 | 5 |
| Q6 | Teaching of interdisciplinary skills (e.g., STEAM, critical thinking, problem solving, etc.) | 1 | 2 | 3 | 4 | 5 |
| Q7 | IT application in teaching | 1 | 2 | 3 | 4 | 5 |
| Q8 | Student behavior and classroom management | 1 | 2 | 3 | 4 | 5 |
| Q9 | Student development and academic evaluation | 1 | 2 | 3 | 4 | 5 |
| Q10 | Helping students make a good transition between middle and high schools | 1 | 2 | 3 | 4 | 5 |

*2.3. Models*

Based on the data structure of student data nested in teacher data, a duo-tier teacher-student model can be developed to estimate the PSTP effects on student performance in Chinese, mathematics, and chemistry. The measurement model is shown below.

$$\text{Level I}: Q_{ij} = \beta_{0j} + \beta_{1j}Q_{ij-1} + \beta_{2j}X_{ijyear} + \beta_{3j}X_{ijtrack} + \gamma_{ij}$$

$$\text{Level II}: \ \beta_{0j} = \gamma_{00} + \gamma_{01}M_j + \gamma_{02}P_j + \mu_{0j}, \beta_{1j} = \gamma_{10}, \beta_{2j} = \gamma_{20}, \beta_{3j} = \gamma_{30}$$

where tier $I$ is an estimate of students $Q_{ij}$, $Q_{ij}$ is the exit score of student $i$ taught by teacher $j$, $Q_{ij-1}$ is the student's baseline score, $X_{ijyear}$ is the student's graduation year, $X_{ijtrack}$ is the liberal arts or sciences the student studied, $\gamma_{ij}$ is the residual, and $\beta_{0j}$ denotes that a random intercept is used at the teacher level. Tier II is an estimation of $\beta_{0j}$, $M_j$ denotes the pre-service preparation of the jth teacher, $P_j$ denotes the personal characteristic variable of the jth teacher, $\gamma_{00}$ is a constant term, and $\mu_{0j}$ is a residual term.

In the estimation of the measurement model, the teacher tier was used with the methods of a random intercept and fixed slope. To exclude the interference factors at the school level, the school-fixed effect was considered in the estimation of the model. PSTP variables were replaced by the standardized values of the same subject in the multi-tier linear regression.

## 3. Results

### 3.1. Descriptive Statistical Analysis of Pre-Service Teacher Preparation Scale

Since some teachers responded to the pre-service preparation scale with missing values, the descriptive statistics were analyzed for each question after missing values were removed, with the sample size, mean, standard deviation, skewness, and kurtosis reported (Table 2). Of the 10 questions on the pre-service preparation scale, Q1 regarding "Knowledge and understanding of the subject taught" has the highest mean value, which indicates to some extent that teachers have good subject knowledge and understanding before entering the profession. Q6 regarding "Interdisciplinary skill teaching" (e.g., STEAM, critical thinking, problem-solving, etc.) has the smallest mean value, explaining that the PSTP for teaching capabilities for interdisciplinary skills needs to be improved. It could also be observed that the mean values of Q1 to Q4 are significantly larger than those of Q5 to Q10. There may be differences in the latent variables measured in the two parts of the questions that require further analysis.

**Table 2.** Descriptive Statistical Analysis of the Pre-Service Teacher Preparation Scale.

| No. | Sample Size | Mean | Standard Deviation | Skewness | Kurtosis |
|-----|-------------|------|--------------------|----------|----------|
| Q1  | 501 | 3.908 | 0.881 | −0.471 | 2.923 |
| Q2  | 501 | 3.665 | 0.957 | −0.413 | 3.053 |
| Q3  | 501 | 3.699 | 0.916 | −0.339 | 3.017 |
| Q4  | 501 | 3.615 | 0.951 | −0.365 | 3.087 |
| Q5  | 501 | 3.020 | 1.168 | −0.182 | 2.414 |
| Q6  | 501 | 2.547 | 1.231 | 0.249 | 2.136 |
| Q7  | 501 | 2.842 | 1.180 | −0.058 | 2.254 |
| Q8  | 501 | 3.267 | 1.114 | −0.237 | 2.592 |
| Q9  | 501 | 2.990 | 1.207 | −0.158 | 2.266 |
| Q10 | 501 | 2.834 | 1.332 | 0.005 | 1.896 |

### 3.2. Exploratory Factor Analysis of Pre-service Teacher Preparation

It may be biased that a subjective choice is made to split a full scale into subscales given the large number of components included in the PSTP scale. Consequently, we used the exploratory factor analysis approach to analyze the PSTP scale. Through the principal component analysis (PCA), we could know that the overall KMO (Kaiser–Meyer–Olkin) value of the scale was equal to 0.924, which was greater than 0.7. The *p*-value of Bartlett's sphericity test was less than 0.05, indicating that the information overlapping between questions was high and suitable for the factor analysis. Table 3 reports the eigenvalues and variance contribution rate of the factor analysis in which the eigenvalues of common factor 1 and common factor 2 are greater than 1, and the eigenvalues of the remaining common factors are less than 1. Meanwhile, the variance contribution rates of common

factor 1 and common factor 2 are 65.015% and 12.901%, respectively, indicating that two common factors are extracted to replace 77.916% of the information of the original scale. By analyzing the eigenvalues and the variance contribution rates, it could be confirmed that the scale was suitable for extracting two common factors with the pre-service preparation scale split into two subscales.

**Table 3.** Eigenvalues and Variance Contribution Rates of the Pre-service Teacher Preparation Scale.

| Factor | Initial Eigenvalue | | | Sum of Squared Rotating Loads | | |
|---|---|---|---|---|---|---|
| | Total | Percentage of Variance | Cumulative Percentage | Total | Percentage of Variance | Cumulative Percentage |
| 1 | 6.501 | 65.015 | 65.015 | 4.110 | 41.098 | 41.098 |
| 2 | 1.290 | 12.901 | 77.916 | 3.682 | 36.818 | 77.916 |
| 3 | 0.577 | 5.774 | 83.690 | | | |
| 4 | 0.395 | 3.953 | 87.643 | | | |
| 5 | 0.319 | 3.192 | 90.835 | | | |
| 6 | 0.284 | 2.840 | 93.675 | | | |
| 7 | 0.229 | 2.290 | 95.965 | | | |
| 8 | 0.160 | 1.599 | 97.564 | | | |
| 9 | 0.140 | 1.402 | 98.966 | | | |
| 10 | 0.103 | 1.034 | 100.000 | | | |

The factor load array was rotated using the Kaiser standardized maximum variance method to further determine the measured question items for each subscale. The rotated factor load array is shown in Table 4. For the rotated factor load array, the main focus was on the loads of each item by the factors and the larger loads could be grouped under the common factor. By analyzing the factor load of each question, it could be found that loads of Q1 to Q4 were large by common factor 1 and the factor loads of all questions were greater than 0.6. According to the connotation covered by the common factors, we could name common factor 1 as content knowledge. Loads of Q5 to Q10 were large by common factor 2 and the factor loads of all questions were greater than 0.6. We could name common factor 2 as pedagogical content knowledge according to Table 1.

**Table 4.** Rotated Factor Load Array for Pre-service Teacher Preparation Scale.

| No. | Common Factor 1 | Common Factor 2 |
|---|---|---|
| Q1 | 0.844 | 0.218 |
| Q2 | 0.880 | 0.336 |
| Q3 | 0.887 | 0.328 |
| Q4 | 0.852 | 0.384 |
| Q5 | 0.437 | 0.747 |
| Q6 | 0.219 | 0.808 |
| Q7 | 0.201 | 0.753 |
| Q8 | 0.463 | 0.717 |
| Q9 | 0.331 | 0.831 |
| Q10 | 0.281 | 0.844 |

The Cronbach's alpha for the content knowledge subscale was 0.943 by further calculating the scale reliability. The Cronbach's alpha for the pedagogical content knowledge scale was 0.921. The reliability of the two subscales was good.

*3.3. Descriptive Statistical Analysis of Content Knowledge and Pedagogical Content Knowledge*

Table 5 reports the sample size (N), mean (Mean), standard deviation (St. Dev), minimum (min), maximum (max), skewness, and kurtosis for both content knowledge and pedagogical content knowledge. For the overall sample, the sample size was 501 teachers. The mean of content knowledge in the two PSTP subscales was 14.886, which was higher

than the median, while that of pedagogical content knowledge was 17.501, which was less than the median. It indicates that teachers were better prepared for knowledge in the PSTP self-assessment, while their preparation for teaching ability was less adequate than content knowledge. Likewise, the standard deviation of pedagogical content knowledge was large, reflecting the high discrete of the pedagogical content knowledge of the sample teachers.

**Table 5.** Descriptive Statistical Analysis of Teacher Behavior and School Support.

| | Pre-Service Teacher Preparation | N | Mean | St. Dev | Min | Max | Skewness | Kurtosis |
|---|---|---|---|---|---|---|---|---|
| Overall | Content knowledge | 501 | 14.886 | 3.425 | 4 | 20 | −0.291 | 3.042 |
| | Pedagogical content knowledge | 501 | 17.501 | 6.134 | 6 | 30 | 0.111 | 2.339 |
| Chinese | Content knowledge | 178 | 14.399 | 3.524 | 4 | 20 | −0.431 | 3.391 |
| | Pedagogical content knowledge | 178 | 17.152 | 5.97 | 6 | 30 | 0.217 | 2.452 |
| Mathematics | Content knowledge | 199 | 15.357 | 3.462 | 4 | 20 | −0.313 | 2.687 |
| | Pedagogical content knowledge | 199 | 18.111 | 6.294 | 6 | 30 | 0.051 | 2.232 |
| Chemistry | Content knowledge | 124 | 14.831 | 3.131 | 5 | 20 | 0.031 | 2.789 |
| | Pedagogical content knowledge | 124 | 17.024 | 6.073 | 6 | 30 | 0.032 | 2.365 |

The results of the descriptive statistics were further analyzed in three subjects: Chinese, mathematics, and chemistry. In the sample of Chinese teachers, the valid sample size for the pre-service preparation scale was 178, and the Chinese teachers' content knowledge and pedagogical content knowledge were below the overall level. In the sample of mathematics teachers, with a valid sample size of 199 for the pre-service preparation scale, both teacher content knowledge and pedagogical content knowledge in mathematics were above average. In the sample of chemistry teachers, with a valid sample size of 124 for the pre-service preparation scale, the chemistry teachers' content knowledge and pedagogical content knowledge were lower than the overall level but higher than those of the Chinese teachers.

By subject, the results of the descriptive analysis revealed differences in teacher behaviors. If not by subject, the overall impact of various teacher behaviors on student performance may be estimated with biased results.

### 3.4. Effects of Pre-Service Teacher Preparation on Student Performance by Subject

After controlling for individual students' characteristics, school fixed effect, and the teachers' characteristic variables (including gender, academic qualification, whether they graduated from normal universities, whether they were holding officially approved positions/bianzhi, years of teaching, and professional ranks), this study estimated the PSTP effects on student performance using multilayer linear regression, as shown in Table 6.

Regarding Chinese, column (1) shows the PSTP effects on student performance, and content knowledge has a significantly negative effect on student performance ($p < 0.05$). Each standard deviation increase in the content knowledge of Chinese teachers is associated with a significant decrease of 0.043 standard deviations in student performance. Pedagogical content knowledge has a significantly positive effect on student performance ($p < 0.05$). For every standard deviation increase in Chinese teachers' pedagogical content knowledge, the student performance is significantly enhanced by 0.053 standard deviations. In the case of Chinese, teachers' content knowledge related to subject knowledge, pedagogy, and teaching methods learned in their pre-graduation specialized programs do not contribute to student performance, even in reverse. In contrast, teachers' pre-graduation pedagogical content knowledge concerning interdisciplinary skills teaching, applied information technology teaching, and student development and assessment significantly contribute to student performance. Considering that 91.3% of the sample Chinese teachers graduated

from normal schools, normal universities should focus more on Chinese-related prospective teachers' pedagogical content knowledge, such as interdisciplinary skill teaching in training.

**Table 6.** Effects of Teacher Behavior on Student Performance.

| | (1) Chinese | (2) Mathematics | (3) Chemistry |
|---|---|---|---|
| Senior high school entrance examination results | 0.243 *** | 0.277 *** | 0.232 *** |
| | (0.009) | (0.008) | (0.010) |
| Content knowledge | −0.043 ** | −0.019 | 0.020 |
| | (0.021) | (0.023) | (0.030) |
| Pedagogical content knowledge | 0.053 ** | −0.001 | −0.050 * |
| | (0.022) | (0.022) | (0.028) |
| Teachers' characteristic variables | Control | Control | Control |
| Liberal arts and sciences | Control | Control | - |
| Year of graduation | Control | Control | Control |
| School fixed effects | Control | Control | Control |
| Constant term | 0.494 * | 0.393 | 0.647 * |
| | (0.283) | (0.251) | (0.369) |
| var(_cons) | 0.021 | 0.029 | 0.027 |
| | (0.003) | (0.004) | (0.005) |
| var(Residual) | 0.488 | 0.379 | 0.424 |
| | (0.007) | (0.005) | (0.007) |
| ICC | 0.041 | 0.071 | 0.060 |
| | (0.006) | (0.009) | (0.010) |
| Sample size | 10,824 | 11,609 | 7739 |

Note: (1) standard deviations are in parentheses; (2) *, **, and *** represent 10%, 5%, and 1% in significance, respectively; (3) teachers' characteristic variables include gender, academic qualification, whether they graduated from normal universities, whether they were holding officially approved positions/bianzhi, years of teaching, and professional ranks; and (4) liberal arts and sciences indicate the discipline chosen by students, and the graduation year is 2016–2019.

With respect to mathematics, column (2) shows the PSTP effects on student performance and both content knowledge and pedagogical content knowledge have a negative but not significant effect on performance. Thus, PSTP in mathematics does not directly significantly influence students' maths performance.

For chemistry, column (3) embodies the PSTP effects on student performance, and content knowledge has a positive but insignificant effect. Pedagogical content knowledge has a significantly negative effect on student performance ($p < 0.1$). For every standard deviation increase in the teachers' pedagogical content knowledge, the student performance in chemistry will decrease by 0.05 standard deviations. In the case of chemistry, teachers' pre-graduation pedagogical content knowledge concerning interdisciplinary skills teaching, applied information technology teaching, and student development and assessment have a significant inverse effect on student performance.

## 4. Discussions

Based on the description of PSTP, this study estimates the PSTP effects on student performance in the three subjects: Chinese, mathematics, and chemistry. First, the results of the empirical study show that PSTP in Chinese could be classified into content knowledge preparation and pedagogical content knowledge preparation. The results can verify Hypothesis I. According to the connotation covered by the common factors, content knowledge preparation includes teachers' knowledge and understanding of the subjects they teach, teaching competencies, general pedagogy, and teaching methods. The pedagogical content knowledge preparation involves teachers' preparation to differentiate instruction for various types of students, interdisciplinary skill teaching, and classroom management skills. The PSTP classification is consistent with Shulman's (1986) content knowledge in teaching [19].

Second, there are differences in PSTP between subjects. The empirical findings can verify Hypothesis II. Based on the results of the descriptive statistical analysis, the content knowledge and pedagogical content knowledge of Chinese teachers are lower than the overall level, while those of the mathematics teachers are higher than the overall average. The chemistry teachers' content knowledge and pedagogical content knowledge are lower than the overall level but higher than those of the Chinese teachers. It may be due to differences in subject knowledge. There also exist differences in the teachers' competencies by subject. The findings of this study are consistent with those of existing studies [26,27,29,33].

Finally, the effect of PSTP on student performance varies across subjects. The results of the empirical study show partial validation for Hypothesis III. In the Chinese subject, PSTP directly affects student performance in which content knowledge significantly negatively influences student performance, and pedagogical content knowledge significantly positively impacts student performance. In mathematics, PSTP does not have a significant effect on student performance. In chemistry, pedagogical content knowledge in PSTP has a significantly negative effect on student performance. According to existing literature, teacher preparation serves as a significant predictor of student performance. The negative relationship between students' performance and teachers' content knowledge may be contrary to the findings of some existing studies. For example, Monk (1994) suggested that teachers' knowledge positively affects student performance [20]. Meanwhile, the significantly positive effect of pedagogical content knowledge on student performance agrees with the findings of relevant research, which confirms the correlation between teacher expertise as well as high-quality teaching and student learning [25]. Given that there may be more teacher discretion in teaching Chinese than in teaching math or chemistry, pedagogical variations in the teaching of Chinese may be more influential on students compared with other subjects. In this way, the findings suggest that Chinese teachers should be more equipped with strong pedagogy than content expertise. Furthermore, it should be noted that the requirements of different subjects for high school students are nationalized, though with minor variations across the provinces, which may be comparable to the US curriculum variation (Daun-Barnett and St. John, 2012; St. John, 2006; St. John and Musoba, 2010) [23–25]. Therefore, the results based on the survey analysis of the Haidian District, Beijing, can be generalized to the nationwide population to a certain extent.

## 5. Limitations

First, post-hoc retrospective data did not allow for rigorous causal inference research. This study was based on the administration data of student performance and teacher questionnaires. The teacher data were only questionnaire data collected at the same time point. Matching the student and teacher data only enabled the analysis of the correlation between individual teacher characteristics and student performance based on cross-sectional data. If feasible, the best way is to track the evaluation and collect baseline data, process data, and outcome data, to compose longitudinal data for causal inference. Second, it should be noted that the survey measures teachers' perception that they are prepared in terms of content knowledge and pedagogical content knowledge, which may be accurately related to their actual perception or there may be biases. Third, the detailed student information was not collected and controlled. Since the 2016–2019 senior high school graduates involved in this study had left school, it was difficult to contact them to do the questionnaires again. If feasible, variables such as students' characteristics and parental background should be controlled in the model, making the estimates more accurate.

## 6. Conclusions

First, PSTP can be divided into content knowledge preparation and pedagogical content knowledge preparation.

Second, there are differences in PSTP by subject. Chinese and chemistry teachers' pre-service preparation is below average, while mathematics teachers' pre-service preparation is above average.

Third, the impact of PSTP on student performance varies across subjects. Chinese teachers' content knowledge preparation significantly negatively influences student performance in Chinese, while their pedagogical content knowledge preparation significantly positively impacts student performance in Chinese. Chemistry teachers' pedagogical content knowledge preparation has a significant negative effect on students' chemistry achievement.

In general, the research on the impact of teacher preparation on student achievement can provide guidance to better develop the criteria for teacher selection. Meanwhile, the empirical results reveal differences in teachers' pre-service preparation in different subjects and the impact on student performance could be inconsistent. Therefore, we should further devise a more accurate teacher training system by subject and give teachers proper training to promote their development as well as the innovation of teaching practice.

**Author Contributions:** Conceptualization, X.L. and L.C.; methodology, X.L.; data curation, X.L. and W.G.; writing—original draft preparation, X.L. and W.G.; writing—review and editing, X.L. and W.G.; funding acquisition, L.C. All authors have read and agreed to the published version of the manuscript.

**Funding:** This paper was funded by the International Joint Research Project of Faculty of Education, Beijing Normal University (ICER201908).

**Institutional Review Board Statement:** Ethical review and approval were waived for this study as we did not collect any information that could identify the participants during the data collection process. Therefore, according to the ethical review regulations, such research does not require ethical review.

**Informed Consent Statement:** Written informed consent has been obtained from the subjects involved in the study.

**Data Availability Statement:** The data that support the findings of this study will be made available from the authors upon reasonable request.

**Acknowledgments:** The authors would like to thank Yi Wei of the China Institute for Educational Finance Research at Peking University for her data support.

**Conflicts of Interest:** The authors declare no conflict of interest.

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
