# Peer review of "Does Pre-Service Teacher Preparation Affect Students’ Academic Performance? Evidence from China"

_education, doi:10.3390/educsci13010069_

Round 1

Reviewer 1 Report

I appreciate you letting me read and share this fascinating paper and topic. It is abundantly evident that the authors did a good job, not only in selecting such a topic, but also in formatting, structuring, and writing up the work.

After a few changes, I believe the manuscript might be published.

1. I believe the introduction may benefit from some expansion by including more contemporary studies and published works of literature.

2. After reading the whole methods section, I was unable to determine whether the researchers employed primary or secondary data to compile their findings; the distinction was not made clear to me. Would you mind elaborating on it for me? If you employed secondary sources of information, it is imperative that you make this point crystal apparent.

3. Because the list of references is so short, I believe that this problem will be resolved once some studies are included in both the introduction and the discussion of the topic.

Once more, I want to thank you for sharing this paper, and I want to wish you the best of luck in the upcoming revision.

Author Response

Thank you, please see attachment

Reviewer 2 Report

This manuscript highlights the impact of pre-service teacher preparation (PSTP) on student performance. The research sample consists of 542 teachers and 39,894 high school students in China. The study was quantitative, consisting of an instrument for PSTP (a 10-question questionnaire) and student-standardized Chinese, mathematics, and chemistry test results of the college entrance exam. Then exploratory factor analysis was used to analyze the data. The PSTP was first divided into content knowledge and pedagogical content knowledge preparation. Next, the researchers used a multilevel linear model to evaluate the impact of PSTP on the performance of high school students.

It is seen that the author/s have done a lot of work and gathered a lot of data, yet the manuscript needs major revisions.

To start with, the author/s needs to follow the Education Science journals guidelines for referencing and citations. There are technical issues that need to be addressed throughout the manuscript. The manuscript needs a deep English revision (grammar and spelling) and, at some points, sentence rewriting and restructuring. Furthermore, the manuscript is not scientifically backed up with relevant references. It needs newer and more relevant referencing throughout the manuscript.

I understand that the manuscript's goal is to provide more information on the impact of pre-service teacher preparation (PSTP) on student performance that would support teacher preparation before an educational reform in China. However, the research presented in this paper needs to reflect and give more information about the current PSTP preparation programs. The researchers are discussing pre-service teacher preparation yet are questioning current in-service teachers. The researchers should present demographic information on the teachers, such as previous teacher preparation, years of experience, PD programs offered or taken, and other relevant information. One huge gap in the manuscript is that the definition of CK, PCK, and PK is insufficient, and throughout, it gives the reader the impression that the researchers do not have a clear understanding of the concept. Great that Shulman is referenced, but it needs a clear interpretation of Shulman's explanation of CK, PCK, and PK. By addressing relevant references, the researchers can build a framework for the relationship between teacher preparation and student achievement in China. Additionally, this might clarify the pre-service teacher preparation as referred by the manuscript to the in-service teacher preparation. The terms subject matter, subject knowledge, and content knowledge are used interchangeably, which confuses the reader.

In lines 118-120, the researchers mention globally used questionnaires. Which global questionnaires are they referring to? Please specify more clearly.

In section 2.2. Measures page 3, the last paragraph is quite confusing as the manuscript is pairing teachers with students while questioning pre-service teachers; see lines 140-141.

In line 48, Most studies need to be referenced. Missing reference on line number 49-50. Throughout the manuscripts, the references are mixed between the APA style and the style used by the journal. Abbreviations such as KMO need to be defined, line number 189. How is the information presented in lines 227-229 related to the current study?

The manuscript misses a thorough literature review on reflection in the introduction and throughout.

Throughout the manuscript, the author/s provides structured information, yet they are described and not critically analyzed, and it can be better presented. Methods, Results, and Discussions are weak and not backed up with literature. The study's methodology is not specified, even though it was understood. The author/s is recommended to hunt and cast out all of the unnecessary words/ or paragraphs that might slow down the reader. Table 1 needs a better representation, or a survey could be placed on the supplementary materials. It is still confusing and requires reasoning on how and why the authors considered Q1-4 as content knowledge and Q5-10 as PCK; see line numbers 211 – 214. Please check the content in lines 296-300 and back it up with the literature.

The conclusion remarks need work. How this idea is presented will contribute to researchers and educators or/and the educational reform in China. Various questions can be raised based on the systematic analysis and conclusion remarks.

Based on preliminary difficulties in comprehending the manuscript's message, I suggest that the manuscript be revised and Resubmitted.

Best regards,

Author Response

Thank you, please see attachment

Round 2

Reviewer 2 Report

Dear authors,

You have done quite a lot of work and have improved the manuscript.

I suggest that you further clarify within the manuscript that the current study's analysis of in-service teachers and student performance will help the pre-service teacher preparation educational reforms in China. I raise this concern since there is no information about the teachers. What is the experience of the in-service teachers? Have they continued with PD programs or other PD courses? Furthermore, because you continuously refer to CK and PCK, a clear explanation in the context is necessary, especially since that is the basis of how you make your analysis and conclusions.

Additionally, please consider the technical issues in the manuscript, i.e., provide some relevant references after “Most studies….”

Good luck!
